# Clinical Significance of Nutritional Status, Inflammation, and Body Composition in Elderly Hemodialysis Patients—A Case–Control Study

**DOI:** 10.3390/nu15245036

**Published:** 2023-12-08

**Authors:** Mar Ruperto, Guillermina Barril

**Affiliations:** 1Department of Pharmaceutical & Health Sciences, School of Pharmacy, Universidad San Pablo-CEU, CEU Universities, Urbanización Montepríncipe, 28660 Alcorcón, Madrid, Spain; 2Nephrology Department, Hospital Universitario La Princesa, C/Diego de León 62, 28006 Madrid, Spain

**Keywords:** aging, bioimpedance analysis, body composition, case–control study, elderly, geriatric nutritional risk, hemodialysis, inflammation, obesity, older adults, phase angle, sarcopenic obesity, s-albumin

## Abstract

Nutritional and inflammatory disorders are factors that increase the risk of adverse clinical outcomes and mortality in elderly hemodialysis (HD) patients. This study aimed to examine nutritional and inflammation status as well as body composition in older adults on HD compared to matched controls. A case–control study was conducted on 168 older participants (84 HD patients (cases) and 84 controls) age- and sex-matched. Demographic, clinical, anthropometric, and laboratory parameters were collected from medical records. The primary outcome was nutritional status assessment using a combination of nutritional and inflammatory markers along with the geriatric nutritional risk index (GNRI). Sarcopenic obesity (SO) was studied by the combined application of anthropometric measures. Body composition and hydration status were assessed by bioelectrical impedance analysis (BIA). Univariate and multivariate regression analyses were performed to identify nutritional and inflammatory independent risk indicators in elderly HD patients and controls. A significantly high prevalence of nutritional risk measured by the GNRI was found in HD patients (32.1%) compared to controls (6.0%) (*p* < 0.001). Elderly HD patients were overweight and had lower percent arm muscle circumference, phase angle (PA), serum albumin (s-albumin), as well as higher percent extracellular body water (ECW%) and serum C-reactive protein (s-CRP) than controls (all at least, *p* < 0.01). SO was higher in HD patients (15.50%) than in controls (14.30%). By multi-regression analyses, age < 75 years (OR: 0.119; 95%CI: 0.036 to 0.388), ECW% (OR: 1.162; 95%CI: 1.061 to 1.273), PA (OR: 0.099; 95%CI: 0.036 to 0.271), as well as BMI, s-albumin ≥ 3.8 g/dL, and lower s-CRP were independently related between cases and controls (all at least, *p* < 0.05). Elderly HD patients had increased nutritional risk, SO, inflammation, overhydration, and metabolic derangements compared to controls. This study highlights the importance of identifying nutritional risk along with inflammation profile and associated body composition disorders in the nutritional care of elderly HD patients. Further studies are needed to prevent nutritional disorders in elderly HD patients.

## 1. Introduction

The global aging of the population, with special attention to the elderly, favors the development of chronic non-communicable diseases and nutritional disorders. By the mid-century, the world’s population over the age of 65 is expected to increase from 10% in 2022 to 16% in 2050 [1].

Chronic kidney disease (CKD) represents a growing public health problem in the world’s aging population. The number of people receiving renal replacement therapy (RRT) is expected to reach 5439 million dialysis patients by 2030 [2]. The most common aging-related risk factors, diabetes mellitus (DM), arterial hypertension, obesity, and cardiovascular disease (CVD), have been identified as the main causes, leading to end-stage renal disease (ESRD) in those over 70 years of age [3].

Overweight and obesity are common conditions showing a U-shaped association with all-cause mortality, especially in older adults ≥ 65 years [4,5,6], as well as in patients with CKD and/or dialysis [7,8]. Body mass index (BMI) remains the most common adiposity index used in health settings. However, significant physiological changes in body composition occur in the aging process that are not reflected in the BMI measurement. Specifically, reduced muscle mass (MM) and increased fat mass (FM) are related to high abdominal adiposity and sarcopenic obesity (SO), which have been reported to increase the risk of mortality in older adults and HD populations [9,10,11]. In addition, age-related nutritional risk factors such as anorexia, inadequate food intake, and underweight often coexist with multifactorial conditions such as cognitive and functional impairment, malnutrition, frailty, and sarcopenia [12].

Different methods of body composition analysis have been used, such as dual-energy X-ray absorptiometry and computed tomography (CT), but their high cost often limits their routine application in clinical settings. Bioelectrical impedance analysis (BIA) is a portable and non-invasive method used to evaluate both hydration status and body composition. BIA-derived measures such as MM, total body water (TBW), body cell mass (BCM), and phase angle (PA) have been reported as independent risk predictors of mortality in the general population [13,14] and in HD patients [15,16,17,18].

Nutritional disorders are a multidimensional and common health problem in the elderly population, with a negative impact on both comorbidity and mortality [19,20,21]. Nutritional risk has previously been identified in 8.5% of community-dwelling older adults [12] and can be as high as 75.0% of the older population on dialysis [22,23]. In particular, some of the nutritional factors involved, such as accumulation of uraemic toxins, metabolic disturbances (hyperparathyroidism, insulin resistance, metabolic acidosis), as well as different degrees of inflammation and tissue or intravascular congestion, are often associated, alone or together, with CVD and adverse outcomes in HD [24,25,26]. 

The Geriatric Nutritional Risk Index (GNRI) is a composite nutritional screening tool that combines serum albumin (s-albumin) and actual to ideal body weight ratio [27,28]. A low GNRI score is significantly correlated with an increased risk of morbidity and all-cause mortality in non-CKD older adults with different diseases [29,30] and in dialysis patients [31,32,33,34,35,36]. Furthermore, the so-called protein-energy wasting (PEW) syndrome is proposed as the diagnostic entity to identify nutritional compromise in CKD. PEW diagnosis is based on several categories (biochemical markers, body mass, muscle mass, and dietary intake), and the combination of these with inflammation adversely influences the outcome [37]. 

The prevalence of PEW varies between 28.0 and 75.0% [22,23] depending on the criteria used for diagnosis. Thus, this fact underlines the importance of a comprehensive and multifactorial approach in HD patients. Given the marked impact of nutritional disorders in the elderly population, this prompted us to evaluate the association of nutritional and inflammatory risk in HD patients compared to controls. The primary outcome was nutritional status assessment using a combination of nutritional and inflammatory markers along with the GNRI score. The identification of nutritional disorders in aging is of particular interest, especially in elderly HD patients. Therefore, this study aimed to examine nutritional status, and inflammation as well as body composition in older adults on HD compared to age- and sex-matched controls. 

## 2. Materials and Methods

### 2.1. Study Design and Participants

This case–control study was conducted retrospectively in the HD unit of the Hospital Universitario La Princesa (Madrid, Spain) from September 2009 (cases) until December 2020 (controls). Cases were retrospectively recruited from the HD unit, while controls were prospectively selected from the community. The inclusion criteria for case selection were age of over 65 years old and undergoing periodic HD treatment (3 times per week, 4 h per HD session) in the last 12 months. Controls enrolled in the study met the following inclusion criteria: over 65 years of age, without cognitive impairment, functionally independent to perform activities of daily living on their own, and absence of CKD or any related medical condition requiring artificial nutritional support. 

Exclusion criteria for cases and controls were advanced heart disease; chronic lung disease; cirrhosis (stages 3, 4); chronic lung disease; active malignancies or bacterial infections; amputation of a limb or pacemaker users; drug treatment with corticosteroids; hospital admission or surgical operations with impact on nutritional status, inflammation, and body composition in the last 3 months; artificial nutritional support including oral nutritional supplements (ONS), enteral tube feeding, intradialytic parenteral nutrition, or total parenteral nutrition; any additional pathology with a life expectancy lower than 3 months.

Figure 1 shows the flowchart of the study’s recruitment of cases and controls. Of the 220 subjects initially recruited, 25 subjects did not meet the inclusion criteria. A total of 195 subjects were included in the study, of whom 95 subjects were HD patients (cases) and 100 subjects were controls. From the control group, 16 participants were excluded, and 11 elderly HD patients were excluded from the case group. Age-matching between cases and controls was within ±5 years. This case–control study was conducted on 84 HD patients and 84 age- and sex-matched controls. 

The study was carried out following the Declaration of Helsinki and Good Clinical Practice Guidelines and was approved by the local Ethics Committee (approval code number: 5320; 12 September 2023). All participants signed a written informed consent form before starting the study.

### 2.2. Sample Size

A minimum sample size of 68 cases and 68 matched controls was required to detect a minimum odds ratio of 0.2 accepting an alpha risk of 0.05 and a beta risk of 0.2. A lost-to-follow-up proportion of 15.0% was considered in this study. The primary outcome was nutritional status assessment using a combination of nutritional and inflammatory markers along with the GNRI score. Secondary outcomes included SO, as well as inflammation profile, hydration status, and age-related changes in body composition.

### 2.3. Study Protocol and Data Collection

At baseline, sociodemographic variables were collected for all participants, including personal information and clinical data from medical records. The etiology of CKD, type of vascular access (arteriovenous fistula or central venous catheter), and dialysis vintage were checked in the case group. All HD patients were dialyzed regularly by standard HD or online hemodiafiltration (online HDF) for at least 4 h per session, three times per week, using ultrapure water and high-flux biocompatible dialysis membranes. Dialysis adequacy measured by Kt/V urea (single pool) was calculated according to the second-generation Daurgidas equation [38].

### 2.4. Anthropometric Measurements 

Body weight (BW), height, triceps skinfold thickness (SKF), mid-arm muscle circumference (MAMC), and waist circumference (WC) were measured in all participants with standardized methods. Height was measured using a stadiometer (HM 200P Charder^®^) and BW (kg) by an electronic scale (Kern MP^®^). Percent of standard body weight (SBW) was calculated as follows: SBW (%) = (BW/SBW) × 100, where BW was the patient’s body weight and SBW was the standard body weight of older Spanish people of the same sex, height, and age range [39]. Post-dialysis dry BW was used for calculating anthropometric measurements. 

Body mass index (BMI) was calculated as BW (kg) divided by squared height (m^2^). The BMI cut-off point was set at <23 kg/m^2^ to define nutritional risk in elderly HD patients [37] and controls [4]. A BMI range of 25–29.9 kg/m^2^ was set for overweight, and BMI ≥ 30 kg/m^2^ for obesity [40].

SKF measurement was performed by Lange Skin Calipers (Cambridge Instruments, Cambridge, MD, USA) and mid-arm circumference (MAC) was measured at the midpoint of the non-dominant or arteriovenous fistula-free arm. Mid-arm muscle circumference (MAMC) was calculated as follows: MAMC (cm) = MAC (cm) − 0.314 × SKF (mm). The percentages of MAMC (%) and SKF (%) were compared with anthropometric reference values at the 50th percentile for the older Spanish population [39]. WC was measured at the mid-point between the lower border of the rib cage and the iliac crest using a rubber measuring tape according to World Health Organization guidelines [41]. Abdominal adiposity was defined as WC ≥ 102 cm for males and WC ≥ 88 cm for females. Sarcopenic obesity (SO) was evaluated using the WC cut-off points by sex combined with MAMC < 90% according to the consensus diagnostic criteria proposed by ESPEN/EASO [42].

### 2.5. Analysis of Body Composition 

Body composition was evaluated using a bioelectrical impedance analyzer (BIA) device (BIA-101^®^. Akern-RJL Systems, Florence, Italy). BIA measurement was performed in the control group on fasting, while in the case group, it was performed 30 min after the end of the dialysis session. The BIA technique has been validated as a method of body composition analysis for controls [43] and for HD patients [44,45].

BIA-derived measurements such as total body water (TBW), extracellular water (ECW), intracellular water (ICW), and exchangeable Na/K were obtained as a basis for the hydration profile. The BIA parameters used to assess body composition were body cell mass (BCM), fat-free mass (FFM), muscle mass (MM), and fat mass (FM) (expressed in kg or %). Phase angle (PA) was also used to indicate cellularity and cell membrane integrity. All parameters obtained by BIA were calculated using Bodygram Pro v.3 software^®^.

### 2.6. Laboratory Parameters

Blood tests from the control group were collected on fasting conditions, and those from the case group were drawn pre-dialysis on the midweek dialysis day. Blood samples included serum cholesterol (s-cholesterol), s-triglycerides, s-creatinine, s-phosphorous, and protein profile (s-albumin, s-prealbumin, s-transferrin), as well as hemoglobin and total lymphocyte count, which were measured by standard automated analyzers (Abbot, Aeroset^®^, Diamond Diagnosis, Holliston, MA, USA). 

Based on the diagnostic criteria proposed for PEW syndrome in HD patients [37], the cut-off point for s-albumin was at 3.8 g/dL. s-Prealbumin and s-CRP (no-high sensitive method), were analyzed by immunoturbidimetry assays (Roche/Hitachi 904^®^/Model P: ACN 218 Roche Diagnostics, Basel, Switzerland). s-CRP ≥ 1.0 mg/dL was used as a diagnostic criterion for inflammation.

### 2.7. Geriatric Nutritional Risk Index 

Nutritional risk was assessed using the Geriatric Nutritional Risk Index (GNRI), a composite screening tool validated for the elderly population [28] and for HD patients [33,34,35,46]. The GNRI was modified by Yamada et al. [46] and calculated as follows: GNRI = [14.89 × albumin (g/dL)] + [41.7 × (body weight/ideal body weight)]. Ideal body weight in this study was defined as the SBW at the 50th percentile of Spanish older adults of the same sex, height, and age range [39]. Post-dialysis dry BW was utilized in HD patients, whereas fasting BW was used in controls. The GRNI cut-off point < 92 points was applied to assess nutritional risk based on previous studies in Caucasian HD patients [29,30,35].

### 2.8. Statistical Analysis

Inclusion and cleaning of the data in the database, pairing, processing, and statistical analyses were carried out during 2023. Data were analyzed using descriptive statistics as the mean and standard deviation for continuous variables and frequency for categorical variables. Cases and controls were age–sex matched at a ratio of 1:1 for data analyses. Chi-square and Fisher’s exact test were used for categorical variables and Student’s *t*-test for continuous variables. To assess the strength of the association of the variables between both cases and controls, Pearson’s chi-square test was used and represented by the heatmap correlation matrix. Correlation was defined by the correlation coefficient (*r*). The degree of correlation was classified based on the *r* coefficient as very weak (*r*: 0.20–0.39), weak (*r*: 0.40–0.59), moderate (*r*: 0.60–0.79), and strong (*r*: 0.80–1.00) [47]. 

Univariate logistic regression analysis and the corresponding odds ratio (OR) and 95% confidence interval (95%CI) were calculated. By multivariate logistic regression analysis using cases and controls as the dependent variable, only those independent variables or potential risk factors with a *p*-value of 0.10 or less were included in the binary logistic regression model. Forward stepwise regression was applied as a method of analysis by sequentially adding each variable one at a time. To test for potential confounders in the multivariate regression, the direction of association was explored using Pearson’s Chi-square parametric correlations together with collinearity and the change of more than 10% in the coefficient (the OR) of the variable(s) included in the model. Statistical analyses were conducted with Statistical Package for the Social Sciences (SPSS) v. 28.0 (IBM Corp., Armonk, NY, USA) statistical software. 

## 3. Results

### 3.1. Global Data and Comparison between Cases and Controls

Table 1 summarizes the clinical and body composition parameters of the participants in the study. A total of 168 older participants age- and sex-matched completed the study, of whom 84 were HD patients (cases) and 84 were controls, mainly females (52.40%). More than 80% (n = 137) of the participants were aged 75 years old or older (*r*: 65–85 years). DM was higher among cases (*n* = 23 (13.70%)) compared with controls (*n* = 14 (7.60%); *p* = 0.36).

Dialysis vintage was 36.25 ± 34.24 months (*r*: 6–170). Among patients on HD, 42.90% (*n* = 36) were on central venous catheter, and based on HD techniques, 61.90% (*n* = 52) were on standard HD and 38.10% on HDF-online (*n* = 32). Residual urine volume (RUV) was not recorded in this study. The mean Kt/V *urea* was 1.59 ± 0.49.

Mean SBW% and BMI differed significantly among HD patients and controls. Controls were significantly more overweighted and obese (*n* = 67; 79.76%) than HD patients (*n* = 35; 42.86%) (*p* < 0.001). Females had higher WC values ≥ 88 cm (cases: 36.0%; controls: 41.9%; *p* = 0.008) than males with WC ≥ 102 cm (cases: 12.50%; controls: 25.0%; *p* = 0.021). Overweight and obesity combined with an increase in sex-adjusted WC were lower in HD patients (32.14%) than in controls (60.70%). As expected, mean SKF% values were increased in both groups, whereas MAMC% was markedly lower in HD patients compared to that of the controls (*p* < 0.001). SO was found in 13 elderly HD patients (15.50%) and 12 paired-matched controls (14.30%). Similarly, BIA-derived measurements, such as resistance, reactance, and body composition parameters were significantly decreased in HD patients in comparison to controls (Table 1, Figure 2). Exchangeable Na/K differed significantly between the two groups as a hydration parameter. PA ≤ 4° as a marker of cell integrity was only observed in 22 older HD patients (30.66%) and none of the controls. 

Figure 2 shows body composition parameters measured by BIA in elderly HD patients and controls. HD patients had significantly lower mean values of MM% and FM%, while FFM% was found to be higher compared to controls (*p* < 0.001).

Figure 3 shows the pattern of hydration status in HD patients and controls. Low ICW% and BCM% along with an increase in TBW% and ECW% were found in HD patients (at least *p* < 0.01).

Table 2 shows the clinical and laboratory parameters of 164 participants in the study. As expected, the mean concentrations of s-triglycerides, s-creatinine, and s-phosphorous were significantly different among cases and controls (all *p* < 0.001). Elderly HD patients had significantly lower concentrations of s-albumin and s-transferrin, while s-ferritin and s-CRP levels were increased compared to those of matched controls (at least *p* < 0.05). No significant mean differences were found with s-cholesterol, hemoglobin and total lymphocyte count between the two groups. A lower mean GNRI score was observed in HD patients than in controls (*p* < 0.001). 

^#^*p*-Values are based on Chi-square or Student’s *t*-test. s-CRP, serum C-reactive protein; GNRI, geriatric nutritional risk index.

Figure 4 shows the correlation matrix of GNRI with anthropometric, body composition measurements, and laboratory data. GNRI correlated positively in cases and controls with MAMC% (both, *p* < 0.001), BCM (kg) (cases: *p* = 0.034; controls: *p* < 0.001), FM% (both, *p* < 0.001), and negatively with exchangeable Na/K (cases: *p* < 0.001; controls: *p* = 0.128) and s-CRP (cases: *p* = 0.012; controls: *p* = 0.972). GNRI < 92 points was significantly more prevalent in HD patients (*n* = 27; 32.10%) than in controls (*n* = 5; 6.0%) (*p* < 0.001).

The nutritional risk assessed by different parameters and thresholds ranged in HD patients between 32.1% and 54.8%, while it was significantly lower in controls, as shown in Figure 5.

### 3.2. Univariate Conditional Regression Analyses

Table 3 shows that BMI, WC, BIA-derived body composition measures (BCM, FM, FFM, MM) and hydration status parameters (TBW, ECW, ICW), as well as some laboratory biomarkers (s-albumin, s-prealbumin, s-ferritin, s-CRP) were significant and independently associated between HD patients and controls by univariate analyses.

### 3.3. Multivariate Regression Analysis

By multivariate logistic regression analysis, it was shown that age < 75 years (OR: 0.119; 95%CI: 0.036 to 0.388), BMI ≥ 23 kg/m^2^ (OR: 0.169; 95%CI: 0.051 to 0.562), body composition parameters such as ECW% (OR: 1.162; 95%CI: 1.061 to 1.273), PA (OR: 0.099; 95%CI: 0.036 to 0.271), as well as s-albumin ≥ 3.8 g/dL (OR: 0.251; 95%CI: 0.073 to 0.870) and s-CRP < 1 mg/dL (OR: 0.056; 95%CI: 0.013 to 0.235) were independent risk variables in HD patients and their control counterparts (all, at least *p* < 0.05) (Table 4).

## 4. Discussion

This case–control study suggests that elderly HD patients had significantly higher nutritional and inflammatory compromise as well as overhydration and body composition disorders compared to their age- and sex-matched controls.

Aging per se increases the vulnerability to nutritional disorders, especially when accompanied by various comorbidities such as DM, CVD, and CKD in older adults. Nutritional disorders, combined with age-related comorbidities, are causative factors in the onset of CKD and, in turn, promote adverse clinical outcomes in the course of the disease. This prompted us to use a comprehensive clinical approach to explore the nutritional, inflammatory, and body composition status in elderly HD patients compared to matched controls. 

Epidemiological studies [4,5] have suggested that overweight or obese elderly people had a paradoxical improvement in survival if the nadir of the BMI curve was between 24.0 and 30.9 kg/m^2^. This phenomenon, known as the “reverse epidemiology or obesity paradox”, showed that higher BMI had a reverse U-shaped all-cause mortality trend, particularly in older adults ≥ 65 years [4,6], as well as in dialysis patients [7,48]. In this study, more than half of the participants were aged ≥ 75 years (*r*: 65–85 years), with DM, overweight, or obesity disorders often being highly prevalent in this elderly cohort. In fact, multivariate regression analysis demonstrated that a BMI ≥ 23 kg/m^2^ reduced the risk of nutritional disorders by 83.10% (OR: 0.169; 95% CI: 0.051 to 0.52; *p* = 0.004) (Table 4). 

Notwithstanding, a high BMI’s protective effect on survival has been limited to patients with normal or increased MM [13,49]. A pooled meta-analysis involving more than 86,285 elderly people concluded that the prevalence of SO was 15.0% when using MM measurements showing that SO affects more than one in ten older adults worldwide [11]. Particularly in this HD cohort, a reduction in MM% and MAMC < 90% was observed (Figure 2 and Figure 5), related to aging, muscle atrophy, obesity, and increased sex-adjusted WC. These findings are consistent with the diagnosis of SO according to the consensus criteria proposed by EASO/ESPEN [42]. Subsequent sub-analysis of our data showed in a non-significant manner SO in both HD patients (15.50%) and controls (14.30%). 

In the aging process from the sixth decade onwards, physiological changes in body composition occur because of an increase in the proportion of body fat to the detriment of MM. SO has been associated with cardiometabolic disturbances, increased risk of falls and fractures, frailty, and mortality in the elderly non-CKD and dialysis population [50]. Moreover, a retrospective study in elderly HD patients [10] showed that SO was significantly associated with increased abdominal myosteatosis by a CT scan, which is a strong risk factor for mortality in adjusted analyses. As in the aforementioned studies [10,11], whose prevalence of SO ranged from 15.0% to 19.0%, our results are in line with those studies in elderly HD patients.

Overhydration is considered a strong predictor of CV events, poor prognosis, and mortality in HD patients [16,18,25,51]. Numerous studies in dialysis patients [18,25,51], have shown that overhydration varies between 56.50% and 73.10%, depending on the diagnostic criterion and method used. In this study, BIA-derived nutritional and hydration indicators such as TBW%, ECW%, ICW%, and BCM% showed significant differences between HD patients and controls (Figure 3). More specifically, HD patients tended to be overhydrated, showing intercompartmental shifts between the ICW and ECW (Table 1 and Figure 3), as well as higher exchangeable Na/K values than in controls. These results should be interpreted considering that residual urine volume was not recorded in this study. In this regard, factors such as older age, female sex, left ventricular hypertrophy, as well as the type of HD technique and time on dialysis were significantly associated with a more pronounced decrease in residual urine volume [52,53,54,55]. Furthermore, this fact, as described in previous studies [56,57], impaired intra-dialysis volume removal, such as the clearance of protein-bound solutes and middle molecules during the dialysis procedure. Additionally, multivariate regression analysis revealed that ECW% increased the risk of overhydration by 1.162-fold (OR: 1.162; 95%CI: 1.061–1.272; *p* < 0.001) (Table 4). In particular, both adiposity and elevated TBW values and/or ECW expansion can lead to an overestimation of body composition measurements, as seen in this study with FFM%. As previously reported, FFM may be higher in obese and older adults and overhydrated HD patients [16,18,58]. In addition, overhydration was also positively related to vascular refilling dysfunction, immune-mediated inflammation, and hypoalbuminemia encompassing underlying nutritional derangements [59]. 

Nutritional disorders are related to age-associated comorbidities and are significant mortality risk factors in older adults and HD patients [20,23]. In our study, nutritional risk assessed by various clinical and nutritional parameters (SBW, MAMC, BMI, s-albumin, and s-CRP) showed wide disparities among HD patients and controls (Figure 5). Low mean BCM and PA values were significantly decreased in HD patients (Table 1, Figure 3). BCM is a sensitive indicator of cellular energy exchange, whereas PA is a marker of cell membrane integrity, both independent and significant nutritional indicators of worse prognosis and mortality in older adults and dialysis patients [15,60]. Interestingly, sex-adjusted mean PA values were related to cohort aging and were significant independent risk indicators of better nutritional status by univariate and multivariate regression analyses (Table 3 and Table 4). Nevertheless, more than half of the elderly HD patients had muscle wasting, and hypoalbuminemia were also inflamed. Thus, according to the PEW diagnostic criteria [37], the combined use of s-albumin < 3.8 g/dL with elevated s-CRP was 19.10% in this study, while none of the controls met the above criteria.

Chronic low-grade inflammation is a condition that commonly accompanies nutritional disorders, often coexisting with tissue overhydration and excess adiposity, as observed in this study. It should be highlighted that low s-albumin may reflect not only a nutritional disorder but also inflammation and overhydration. A prior study [61] showed that haemodilution hypoalbuminemia is associated with excess TBW and that the excess fluid is not equally localized in the intravascular or extravascular space, as in hypoalbuminemia the oncotic pressure is reduced. In contrast, the combined use of s-albumin ≥ 3.8 g/dL and s-CRP < 1 mg/dL was shown to significantly decrease the risk of nutritional disorders by multivariate logistic regression analysis in our study (Table 4). 

S-prealbumin, a half-life, non-volume-dependent marker of excess fluid, is an earlier indicator of nutritional and inflammatory disorders. In this study, s-prealbumin < 30 mg/dL and s-CRP ≥ 1 mg/dL were found in 30.95% of HD patients, whereas the proportion was only 7.10% in controls.A retrospective study of 798 HD patients [62], reported that a 6-month fall in s-prealbumin was independently associated with an increased risk of death at a 5-year follow-up. The results of previous studies [24,61,62] and the findings of our study suggested that both s-albumin and s-prealbumin are influenced by non-nutritional factors and therefore should be assessed together with inflammatory and hydration parameters in dialysis patients. Previously, we reported that the utilization of s-albumin and s-prealbumin along with s-CRP added predictive value to the nutritional diagnosis in CKD [63] and was also an independent mortality risk factor in HD patients [18,64,65]. 

Additionally, in this study, nutritional risk was screened by the GNRI. The GNRI has been reported to be a simple, objective, and non-invasive composite nutritional risk tool in different disease conditions [29,30,31,32,33,34,35,36]. In this study, the GNRI was positively associated with MAMC%, BCM (kg), FM%, and s-prealbumin and negatively with inflammatory markers such as s-CRP and exchangeable Na/K (Figure 4). The mean GNRI was within the normal range, significantly lower in HD patients than in older adults (Table 2). 

Low GNRI values (<92.0 points) were identified as risk factors for hospitalization [30], infection-related mortality [32], and stroke during 10 years of follow-up [31], as well as all-cause mortality [35,36]. The prevalence of nutritional risk in this cohort as measured by the GNRI was 32.10% in elderly HD patients and only 6.0% in controls. These results are consistent with those of some published studies [31,32,34,35,66,67] but differ from others [68,69]. 

This study has some strengths and limitations. Firstly, to the best of our knowledge, this is the first case–control study in elderly HD patients to compare nutritional risk and inflammation status, as well as body composition measurements, with age- and sex-matched controls. Nevertheless, our results do not necessarily imply causality, and may be biased by other biomarkers not tested in this elderly cohort. Secondly, BMI is a global indicator of adiposity that does not take into account age-related changes in body composition (e.g., increased fat mass) as well as abdominal adiposity or SO in the elderly population. Particularly, in this study, BMI together with WC and additional anthropometric and body composition measurements were used for diagnosing abdominal adiposity and SO. Thirdly, skeletal muscle strength (e.g., handgrip strength, knee extensor strength, or chair-stand test) was not recorded for diagnosing SO. Instead, SO was diagnosed by the conjoint use of sex-adjusted cut-off points of increased WC and decreased MAMC, using the diagnostic criteria recommended by the EASO/ESPEN consensus [42]. Fourth, BIA-derived measurements were evaluated at a single time point. However, our study provided a comprehensive approach to BIA measurements together with nutritional and inflammatory markers such as s-albumin, and s-CRP as recognized risk indicators for nutritional disorders in dialysis. Fifthly, residual diuresis was not measured, which is a partially limiting factor in the interpretation of clinical results. However, in this cohort of prevalent elderly on HD, hydration status was measured after dialysis using several BIA-derived measures (TBW, ECW, ICW, and exchangeable Na/K) as part of the routine clinical and nutritional follow-up protocol in the HD unit. Sixth, food intake was not recorded in this study, although HD patients received regular nutritional counselling following Kidney Disease Outcomes Quality Initiative (KDOQI) practical guidelines recommendations [70]. Medication treatment was not recorded, but some drugs that could interfere with nutritional and inflammatory status (e.g., ONS, corticosteroids) were considered exclusion criteria in this study. Lastly, the GNRI was used as a nutritional risk tool (<92.0 points in this study), although other studies have applied different cut-off points according to ethnicity, stage of CKD as well as in peritoneal dialysis [23,28,31,33,36,68,69,71,72,73], which may diverge from the findings of our study. 

## 5. Conclusions

In conclusion, this case–control study highlights the importance of identifying nutritional risk along with inflammation profile and associated body composition disorders in the nutritional care of elderly HD patients. The results of the multifocal nutritional approach differed significantly in the HD elderly cohort compared to those of age- and sex-matched controls. One third of elderly HD patients had nutritional disorders, while overweight and obesity, along with increased abdominal adiposity and SO, were also especially prevalent in HD patients. s-CRP and ECW% increased the likelihood of nutritional risk, whereas independent biomarkers such as PA and s-albumin ≥ 3.8 g/dL were shown to reduce the risk of nutritional disorders in elderly HD patients. Further multiapproach studies are required for preventing nutritional disorders in elderly HD patients.

## Figures and Tables

**Figure 1 nutrients-15-05036-f001:**
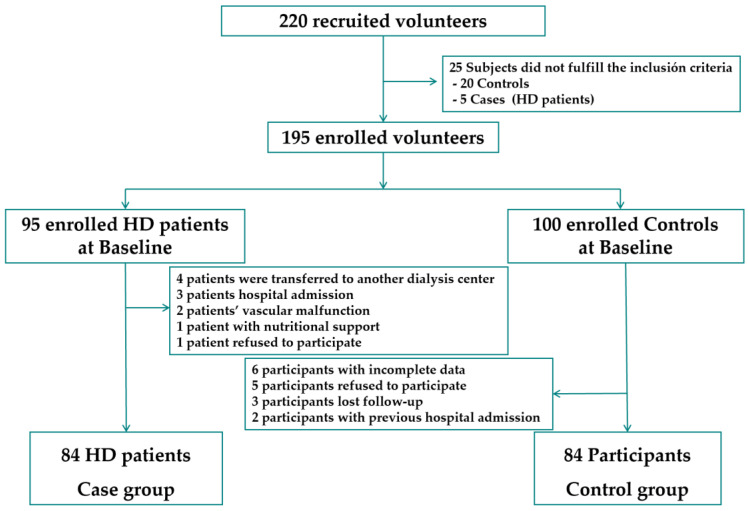
Flowchart of the recruitment and selection of cases and controls in the study. HD, hemodialysis.

**Figure 2 nutrients-15-05036-f002:**
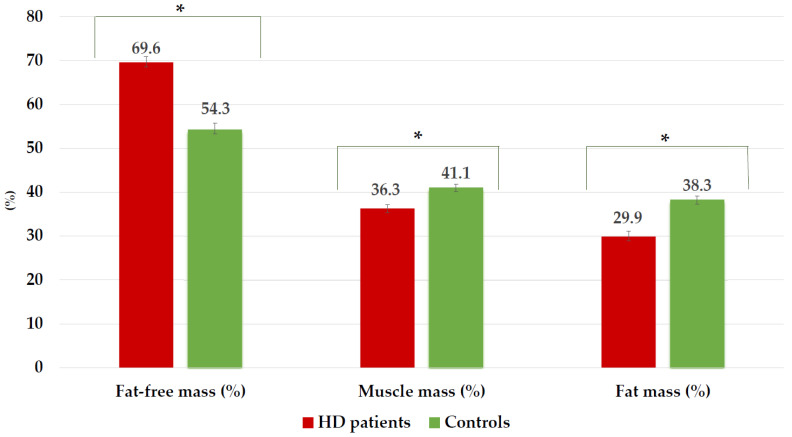
Body composition parameters in hemodialysis patients and controls. Values are expressed in percentages (%). * *p* < 0.001.

**Figure 3 nutrients-15-05036-f003:**
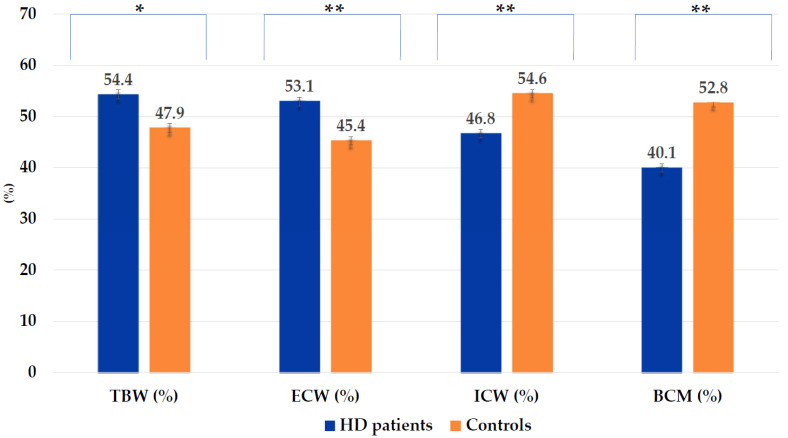
Distribution of hydration status and body cell mass in hemodialysis patients (cases) and controls. BCM%, percentage of body cell mass; ECW%, percentage of extracellular water; ICW%, percentage of intracellular water; TBW%, percentage of total body water. * *p* = 0.006; ** *p* < 0.001.

**Figure 4 nutrients-15-05036-f004:**
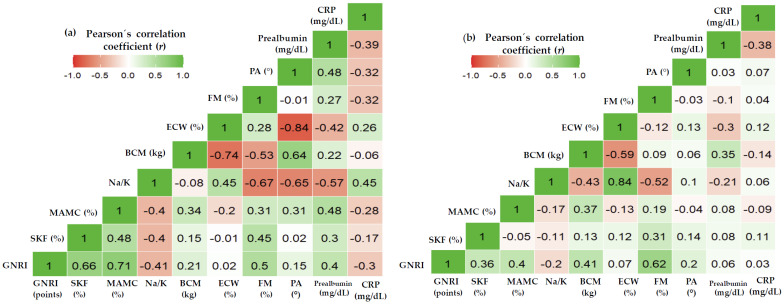
Heatmap correlation matrix of the geriatric nutritional risk index with various anthropometric, body composition, and laboratory parameters in (**a**) cases and (**b**) controls. Correlation is defined by the correlation coefficient (*r*). The degree of correlation was classified based on the *r* coefficient as very weak (*r*: 0.20–0.39), weak (*r*: 0.40–0.59), moderate (*r*: 0.60–0.79), and strong (*r*: 0.80–1.00) [47]. BCM (kg), body cell mass; s-CRP, serum C-reactive protein; ECW%, percentage of extracellular water; FM%, percentage of fat mass; GNRI, geriatric nutritional risk index; MAMC%, percentage of mid-arm-muscle circumference; Na/K, exchangeable Na/K; PA, phase angle; s-Prealbumin, serum prealbumin; SKF%, percentage of triceps skinfold thickness.

**Figure 5 nutrients-15-05036-f005:**
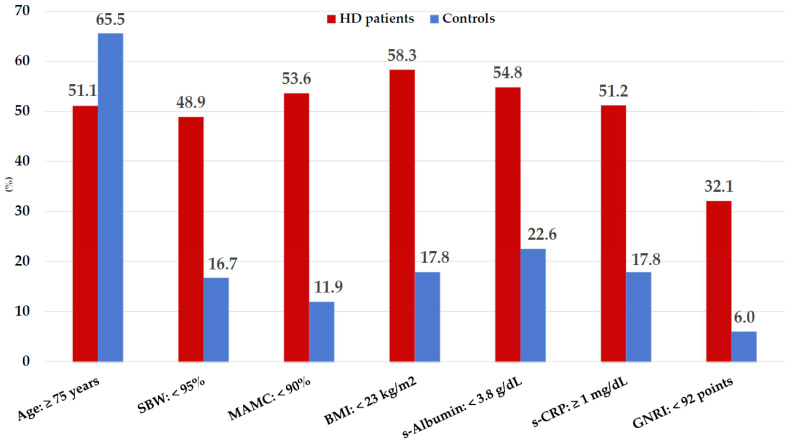
Prevalence of nutritional disorders in older hemodialysis patients and controls based on nutritional marker thresholds proposed for diagnosing and evaluating nutritional status. Values are expressed in percentages (%).

**Table 1 nutrients-15-05036-t001:** Clinical, anthropometric, and body composition parameters of the 168 participants in the study.

Variable	HD Patients (*n* = 84)	Controls (*n* = 84)	*p*-Value
Male, n (%)	40.0 (47.60)	40.0 (47.60)	-
Age (years)	76.40 ± 4.04	77.26 ± 3.75	0.150
DM n; (%)	23.0 (13.70)	14 (7.60)	0.095
BW (kg)	66.72 ± 13.34	66.90 ± 11.27	0.920
SBW (%)	100.42 ± 21.30	119.86 ± 21.84	<0.001
BMI (kg/m^2^)	25.44 ± 4.87	28.54 ± 5.12	<0.001
WC (cm)	96.04 ± 12.97	102.20 ± 11.39	0.001
SKF (%)	121.60 ± 55.64	144.79 ± 45.92	<0.001
MAMC (%)	95.21 ± 10.51	110.33 ± 16.01	<0.001
Resistance (Ω)	506.36 ± 68.18	605.01 ± 72.31	<0.001
Reactance (χ_c_)	40.98 ± 10.02	51.12 ± 6.00	<0.001
FFM (kg)	46.27 ± 9.69	40.76 ± 5.98	0.078
MM (kg)	23.89 ± 5.69	27.11 ± 4.49	<0.001
FM (kg)	20.34 ± 9.95	26.52 ± 8.37	0.008
Exchangeable Na/K	1.32 ± 0.35	0.93 ± 0.18	<0.001
TBW (L)	35.94 ± 6.91	32.37 ± 4.28	0.100
ECW (L)	18.98 ± 3.80	14.68 ± 2.54	<0.001
ICW (L)	16.96 ± 4.46	17.68 ± 3.01	0.010
BCM (kg)	18.52 ± 4.86	21.94 ± 3.89	<0.001
PA (°)	4.26 ± 0.70	5.41 ± 0.90	<0.001

*p*-Values are based on Chi-square or Student’s *t*-tests. BCM, body cell mass; BMI, body mass index; DM, diabetes mellitus; ECW, extracellular water; FFM, fat-free mass; FM, fat mass; ICW, intracellular water; MAMC, mid-arm muscle circumference; MM, muscle mass: PA, phase angle; SBW%, percentage of standard body weight; SKF%, percentage of triceps skinfold thickness; TBW, total body water; WC, waist circumference.

**Table 2 nutrients-15-05036-t002:** Clinical and laboratory parameters of 168 participants in the study ^#^.

Variable	HD Patients (*n* = 84)	Controls (*n* = 84)	*p*-Value
s-Cholesterol (mg/dL)	157.33 ± 42.41	169.97 ± 37.14	0.153
s-Triglycerides (mg/dL)	152.76 ± 85.97	112.64 ± 45.23	<0.001
s-Creatinine (mg/dL)	3.82 ± 1.33	0.95 ± 0.28	<0.001
s-Phosphorous (mg/dL)	4.69 ± 0.77	4.12 ± 0.39	<0.001
s-Albumin (g/dL)	3.73 ± 0.42	3.98 ± 0.30	0.030
s-Prealbumin (mg/dL)	26.40 ± 8.52	18.51 ± 3.56	0.030
s-Transferrin (mg/dL)	171.36 ± 30.97	210.52 ± 25.72	<0.001
s-Ferritin (ηg/mL)	511.86 ± 452.91	106.16 ± 94.46	<0.001
s-CRP (mg/dL)	1.21 ± 0.91	0.64 ± 0.50	<0.001
Hemoglobin (g/dL)	12.10 ± 1.39	12.52 ± 1.36	0.466
Total lymphocyte count (×10^3^/mm^3^)	1361.79 ± 499.41	1947.10 ± 731.92	0.073
GNRI (points)	97.55 ± 11.32	108.47 ± 10.65	<0.001

^#^ *p*-Values are based on Chi-square or Student’s *t*-test. s-CRP, serum C-reactive protein; GNRI, geriatric nutritional risk index.

**Table 3 nutrients-15-05036-t003:** Univariate conditional regression analysis in cases and controls.

Variable	OR	St Error	95%CI	*p*-Value
BMI (kg/m^2^)	0.841	0.037	0.771 to 0.918	<0.001
WC (cm)	0.956	0.014	0.928 to 0.985	0.003
FFM (%)	1.164	0.041	1.086 to 1.247	<0.001
MM (%)	0.902	0.024	0.855 to 0.952	<0.001
FM (%)	0.889	0.023	0.843 to 0.938	<0.001
TBW (%)	1.165	0.041	1.086 to 1.251	<0.001
ECW (%)	1.278	0.063	1.160 to 1.408	<0.001
ICW (%)	0.785	0.038	0.713 to 0.865	<0.001
BCM (%)	0.857	0.026	0.807 to 0.910	<0.001
PA (°)	0.157	0.061	0.073 to 0.337	<0.001
Total cholesterol (mg/dL)	0.987	0.005	0.977 to 0.997	0.011
s-Triglycerides (mg/dL)	1.007	0.003	1.002 to 1.013	0.011
s-Phosphorous (mg/dL)	2.396	0.742	1.305 to 4.398	0.005
s-Albumin (g/dL)	0.341	0.155	0.139 to 0.833	0.018
s-Prealbumin (mg/dL)	0.756	0.077	0.682 to 0.861	<0.001
s-Transferrin (mg/dL)	0.956	0.008	0.938 to 0.973	<0.001
s-Ferritin (ηg/mL)	1.010	0.002	1.006 to 1.014	<0.001
s-CRP (mg/dL)	1.704	0.281	1.233 to 2.355	<0.001
Total lymphocyte count (×10^3^/mm^3^)	0.998	0.003	0.998 to 0.999	<0.001
GNRI (points)	0.881	0.023	0.844 to 0.934	<0.001

*p*-Values are based on univariate regression analysis using cases and controls as dummy variables. BCM, body cell mass; OR, odds ratio; St Error, standard error; 95%CI, 95% confidence interval. BMI, body mass index; ECW, extracellular water; FM, fat mass; FFM, fat-free mass; GNRI, geriatric nutritional index; ICW, intracellular water; MM, muscle mass; PA, phase angle; s-CRP, serum C-reactive protein; TBW, total body water.

**Table 4 nutrients-15-05036-t004:** Multivariate logistic regression analysis in cases and controls.

Variable	OR	St Error	95%CI	*p*-Value
Age (<75 years)	0.119	0.604	0.036 to 0.388	<0.001
BMI (≥23 kg/m^2^)	0.169	0.612	0.051 to 0.562	0.004
ECW (%)	1.162	0.047	1.061 to 1.273	0.001
PA (°)	0.099	0.516	0.036 to 0.271	<0.001
s-Albumin (≥3.8 g/dL)	0.251	0.634	0.073 to 0.870	0.029
s-CRP (<1 mg/dL)	0.056	0.736	0.013 to 0.235	<0.001

*p*-Values are based on multivariate logistic regression analysis using cases and controls as dummy variables. OR, odds ratio; St Error, standard error; 95%CI, 95% confidence interval. BMI, body mass index; ECW%, percentage of extracellular water; PA, phase angle; s-Albumin, serum albumin; s-CRP, serum C-reactive protein.

## Data Availability

The data presented in this study are contained in the article.

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
