# Peer review of "Clinical Significance of Nutritional Status, Inflammation, and Body Composition in Elderly Hemodialysis Patients—A Case–Control Study"

_nutrients, 2023, doi:10.3390/nu15245036_

Round 1
Reviewer 1 Report
Comments and Suggestions for Authors
The manuscript “Clinical Significance of Nutritional Status, Inflammation, and Body Composition in Elderly Hemodialysis Patients. A Case-Control Study” by Ruperto & Barril is a research article which examined nutritional and inflammation status as well as body composition in older adults on elderly hemodialysis (HD) compared to matched controls. The authors found that elderly HD patients had significant sarcopenic obesity including increased nutritional risk, inflammation, overhydration, and metabolic derangements compared to controls. In general, this article is critical in this field and contains essential findings. However, I have several comments before this manuscript is accepted for publication.
1. In bar graphs, all data plots are included if possible. Readers can easily know the distribution of the data.
2. Figure 4. The parameters explaining geriatric nutritional risk index are too small to see. Please revise the figure.
3. Figure 5 should be changed into Table if possible, because there is no statistical data and the information obtained from bar graphs is not useful.
Author Response
Answers to Reviewer #1 comments and suggestions on the article entitled "Clinical Significance of Nutritional Status, Inflammation, and Body Composition in Elderly Hemodialysis Patients. A Case-Control Study" by authors Ruperto M. and, Barril G. (Nutrients code number: 2714962).
The manuscript “Clinical Significance of Nutritional Status, Inflammation, and Body Composition in Elderly Hemodialysis Patients. A Case-Control Study” by Ruperto & Barril is a research article which examined nutritional and inflammation status as well as body composition in older adults on elderly hemodialysis (HD) compared to matched controls. The authors found that elderly HD patients had significant sarcopenic obesity including increased nutritional risk, inflammation, overhydration, and metabolic derangements compared to controls. In general, this article is critical in this field and contains essential findings.
Authors' reply. Many thanks for the review and your valuable comments to improve our manuscript. Suggested changes are highlighted in blue in the second version of the manuscript. Changes highlighted in red are those suggested by the editor.
However, I have several comments before this manuscript is accepted for publication.
1.In bar graphs, all data plots are included if possible. Readers can easily know the distribution of the data.
Authors' reply. Thanks for your comment. The data have been included in the bar charts in Figures 2 and 3, of the version 2 of the manuscript.
2. Figure 4. The parameters explaining geriatric nutritional risk index are too small to see. Please revise the figure.
Authors' reply. Thank you for helping us to improve our manuscript. In the new version of the manuscript, figure 4 has been modified.
3. Figure 5 should be changed into Table if possible, because there is no statistical data and the information obtained from bar graphs is not useful.
Authors' reply. Thank you for your suggestion. Figure 5 summarises globally and visually the prevalence of nutritional disorders by comparing the proportion of each parameter between the case and control cohort. Although no statistics are included, the purpose of including this bar chart is to draw the attention of professionals to the most relevant risk parameters as well as cut-off points for improvement in nutritional assessment and clinical diagnosis in hemodialysis patients.
Thank you again for your comments and suggestions to improve the comprehension of our manuscript.

Reviewer 2 Report
Comments and Suggestions for Authors
This manuscript presents an interesting nephro-nutritional approach in patients with chronic kidney disease (CKD) on hemodialysis (HD). This study highlights the performance of systematic nutritional screening and assessment as well as nutritional care and follow-up as a preventive and therapeutic strategy in CKD patients on HD, through sarcopenic obesity(SO), the composite geriatric nutritional risk index (GNRI), and bioelectrical impedance analysis (BIA) to assess hydration status and body composition.
Indeed, this study puts in their position contradictory elements and data from several published studies of bibliography.
The language seems correct throughout the text and references are in general up-to-date.
However, these important results obtained do not imply causality, biased by unanalyzed biomarkers. A second limitation concerns the global indicator of BMI for adiposity that does not take into account changes in body composition as well as SO especially in age-related CKD on HD. Medication treatment could interfere with nutritional and inflammatory status where disorders present severe factors increasing the risk of mortality in elderly HD patients.
I personally would like to see a revised form of the present study taking into account medication intake and BMI associated with several adipose tissue and CKD biomarkes as FGF-21 and -23, TGF-β1, VEGF-A, MMP-2 and MMP-9, TIMP-1 and -2, ox-LDL, isoprostane-8 and -15, and BNP, in order to better describe and improve this important clinical model.
Author Response
Answers to Reviewer #2 comments and suggestions on the article entitled " Clinical Significance of Nutritional Status, Inflammation, and Body Composition in Elderly Hemodialysis Patients. A Case-Control Study "by authors Ruperto M. and, Barril G. (Nutrients code number: 2714962).
This manuscript presents an interesting nephro-nutritional approach in patients with chronic kidney disease (CKD) on hemodialysis (HD). This study highlights the performance of systematic nutritional screening and assessment as well as nutritional care and follow-up as a preventive and therapeutic strategy in CKD patients on HD, through sarcopenic obesity (SO), the composite geriatric nutritional risk index (GNRI), and bioelectrical impedance analysis (BIA) to assess hydration status and body composition.
Indeed, this study puts in their position contradictory elements and data from several published studies of bibliography.
The language seems correct throughout the text and references are in general up-to-date.
Authors' reply. Thank you very much for the review of the manuscript and the comments and suggestions to improve our paper. Suggested changes are highlighted in blue in the new version of the manuscript. Changes highlighted in red are those suggested by the editor.
1. However, these important results obtained do not imply causality, biased by unanalyzed biomarkers. A second limitation concerns the global indicator of BMI for adiposity that does not take into account changes in body composition as well as SO especially in age-related CKD on HD. Medication treatment could interfere with nutritional and inflammatory status where disorders present severe factors increasing the risk of mortality in elderly HD patients.
Authors' reply. Thanks for your comments. In the second version of the manuscript, following your suggestions and at the request of the editor, we have included the limitation that the results obtained in the study do not imply causality, and there may be undefined bias factors in the study that may affect the results.
In relation to its own limitations as a marker for BMI, in this study we additionally analysed BMI together with anthropometric parameters and body composition analysis by electrical bioimpedance, which considers hydration pattern as well as body composition, including both muscle mass and fat mass according to age and sex.
The medications used were not recorded in this study, but the exclusion criteria of the study considered drugs that could interfere with the nutritional and inflammatory status of these patients. This limitation was included in the discussion section of the study limitations.
2. I personally would like to see a revised form of the present study taking into account medication intake and BMI associated with several adipose tissue and CKD biomarkers as FGF-21 and -23, TGF-β1, VEGF-A, MMP-2 and MMP-9, TIMP-1 and -2, ox-LDL, isoprostane-8 and -15, and BNP, in order to better describe and improve this important clinical model.
Authors' reply. Thank you very much for your comments, although these important biomarkers, which would certainly add value to this study, were not included in the current study. We will consider your suggestions and comments and the use of these biomarkers for future clinical studies.
The authors would like to thank you again for your review and comments that have allowed us to improve the manuscript.

Round 2
Reviewer 2 Report
Comments and Suggestions for Authors
Published in its revised form.